# Roquin Modulates Cardiac Post-Infarct Remodeling via microRNA Stability Control

**DOI:** 10.3390/cells14221748

**Published:** 2025-11-07

**Authors:** Nadja Itani, Rolf Schreckenberg, Rainer Schulz, Peter Bencsik, Peter Ferdinandy, Klaus-Dieter Schlüter

**Affiliations:** 1Physiologisches Institut, Justus-Liebig-Universität Gießen, 35392 Gießen, Germany; nadja.itani@physiologie.med.uni-giessen.de (N.I.); rolf.schreckenberg@physiologie.med.uni-giessen.de (R.S.); rainer.schulz@physiologie.med.uni-giessen.de (R.S.); 2Pharmahungary Group, 6722 Szeged, Hungary; peter.bencsik@pharmahungary.com (P.B.); peter.ferdinandy@pharmahungary.com (P.F.); 3Department of Pharmacology and Pharmacotherapy, University of Szeged, 6722 Szeged, Hungary; 4Department of Pharmacology and Pharmacotherapy, Semmelweis University, 1094 Budapest, Hungary; 5Center for Pharmacology and Drug Research & Development, Semmelweis University, 1085 Budapest, Hungary

**Keywords:** ZBTB20, miR-23b-5p, post-infarct remodeling

## Abstract

Through binding to complementary mRNAs, microRNAs (miRNAs) mediate gene silencing. The stability and half-life of microRNAs are controlled by two isoforms of the RNA-binding protein Roquin. This study aimed at identifying the role of Roquin to miRNA-dependent regulation of the transcriptome in the post-ischemic heart. Both Roquin isoforms are highly conserved between rats and humans and constitutively expressed in cardiomyocytes. In both cell species, hypoxia induces a down-regulation of Roquin-1 and Roquin-2. An integrative miRNA-and-mRNA analysis (MMIA) identified miR-23b-5p as a potential interaction partner of Roquins. The open data bank TargetScan8.0 suggests that the transcription factor ZBTB20 is a potential target of miR-23b-5p. The level of expression of *ZBTB20* correlated with the functional recovery of rat hearts after myocardial infarction. Moreover, the down-regulation of Roquin-2 in AC16 cells by siRNA under normoxic conditions was associated with an up-regulation of miR-23b-5p and a down-regulation of *ZBTB20*. Furthermore, in the case of hypoxia-dependent down-regulation of Roquin, the subsequent down-regulation of *ZBTB20* was reversed with the help of an antagomir against miR-23b-5p. In conclusion, hypoxia-induced down-regulation of the two Roquin isoforms was associated with an increased stability of miR-23b-5p, a Roquin-2-dependent miRNA, which subsequently led to silencing of the transcription factor ZBTB20.

## 1. Introduction

Revascularization of occluded coronary vessels leads to reperfusion injury. The extent and nature of this tissue damage affect the subsequent remodeling of the post-infarcted heart. However, such structural remodeling processes require protein synthesis and degradation. Both processes depend at least in part on transcriptional regulation. A number of specific mechanisms are involved in the transcriptional regulation of genes, such as the regulation of transcription factor expression and activity, epigenetic modifications of genes, control of mRNA half-life, and control of mRNA regulators such as microRNAs (miRNAs). RNA-binding proteins such as Roquin can control the stability and half-life of RNA molecules, including mRNA and miRNAs. However, Roquin expression and function have not yet been addressed in post-infarct remodeling.

Roquins, a family of RNA-binding proteins, contain two family members named Roquin-1 and Roquin-2. Both proteins show a high degree of evolutionary conservation. Their involvement in immune defense in Drosophila has already been demonstrated, but they also play a role in immune regulation in mammals [1,2]. It has been proposed that Roquins originate from the cellular defense of eukaryotic cells against RNA pathogens [3]. As they were first recognized by their effects on immune cells, it is in line with this finding that Roquin deficiency is associated with inflammation in nearly all tissues [4]. In the heart, Roquin deficiency causes T-cell-dependent inflammation [5]. The mechanisms by which Roquins affect transcription, translation, and cell function are complex and contain mRNA deadenylation [6], RNA degradation in connection with Regnase [7], targeting of miRNAs [8], and interference with the Akt signaling pathway [9]. In addition, Roquins also have E3 ubiquitin ligase activity and can degrade proteins [10]. Roquin expression can be inhibited by mucosa-associated lymphoma translocation protein 1 (MALT1) [11,12] and miR-223 [13]. Furthermore, it has already been demonstrated that ischemia in the liver influences the expression of Roquin-1 [14].

Inflammatory processes and altered transcription contribute to post-infarction remodeling, and both processes are influenced by Roquin. However, there are yet no data available about the effect of ischemia and reperfusion on the expression and function of Roquins in the heart. The present study therefore aimed to identify potential cardiac signaling pathways that are influenced by Roquin. To shed light on this topic, the regulation of Roquin expression in post-infarcted cardiac tissue also needs to be examined. We previously analyzed the expression of miRNAs in the reperfused myocardium of rat hearts [15]. We now extended our previous work and used these data and a quantitative mRNA expression approach to perform an integrative miRNA-and-mRNA analysis (MMIA). This approach formed the basis by which potential cardiac targets of Roquin were discovered and led to the identification of miR-23b-5p. Subsequent screening of the TargetScan8.0 database identified the zinc finger and BTB domain containing protein 20 (ZBTB20) as a potential down-stream target of miR-23b-5p. We then used siRNA and antagomir approaches to validate a newly discovered Roquin-2–miR-23-b5p–ZBTB20 cascade that appears to be important for post-ischemic remodeling.

## 2. Materials and Methods

### 2.1. Ethics Statement of Animal Studies

This investigation conforms with the Guide for the Care and Use of Laboratory Animals published by the US National Institute of Health (NIH publication No. 85-23, revised 1996). The animal study protocol (isolation of rat cardiomyocytes) was approved by the Institutional Review Board (or Ethics Committee) of the Justus Liebig University, Giessen, Germany (752_M; 17 August 2023). All other data used in this study from rat tissues were taken from the data set already used before in a different context [15].

### 2.2. Animal Studies

All tissue samples used in this study were obtained from female Wistar-Hannover rats weighing 215–265 g. The sample material was collected as part of an earlier study and further examined in the current work under the new objective. All details regarding surgical procedures, analysis of left ventricular function, and the extraction and processing of miRNA and mRNA are explained in detail in the study that has already been published [15].

### 2.3. Isolation of Rat Ventricular Myocytes and Non-Myocytes

Ventricular heart muscle and non-muscle cells were isolated from male Wistar rats as described before [16]. Briefly, after hearts were excised under deep anesthesia, they were quickly mounted on a Langendorff system and perfused with collagenase-containing buffer. Myocytes and non-myocytes were separated by sedimentation and subsequent resuspension. After isolation and separation, myocytes and non-myocyte cells were quickly frozen in liquid nitrogen and stored until used at −80 °C.

### 2.4. Cell Culture

Human AC16 cardiomyocytes were cultured on 100 mm dishes (Falcon, type 3003, Corning Inc., Corning, NY, USA) in Dulbecco’s Modified Eagle’s Medium (D6429, EMD Millipore Corp., Billerica, MA, USA), added with 12.5% heat-inactivated FBS (35-079-CV, Corning Inc., USA) and under 1% antibiotic–antifungal coverage (30-004-CI, Corning Inc., USA) according to the protocol (#SCC109, EMD Millipore Corp., USA). In this study, all AC16-based experiments were performed with cells that were initially transferred into 60 mm culture dishes (Falcon, type 3002, Corning Inc., USA).

SiRNA transfection was performed according to the Lipofectamine RNAiMAX transfection protocol (life technologies, Carlsbad, CA, USA). For each transfection of a 35 mm culture dish, 4.5 µL of a Lipofectamine transfection reagent was mixed with cold 150 µL Opti-MEM (Thermo fisher, Waltham, MA, USA) and 1.5 µL siRNA for Roquin-1 (HS_KIAA2025, Qiagen, Venlo, The Netherlands), Roquin-2 (HS_MNAB, Qiagen), or for the control (SI03650318, Qiagen) and incubated for 5 min at room temperature. The cell density on the culture dish was 70% confluent for transfection. The cells were then transfected and incubated at 37 °C for up to 48 h.

For the plasmid transfection of the AC16 cell line, the plasmid RC3H1 Human Untagged Clone was used with the vector pCMV6 entry (SC306712, ORIGENE, Rockville, MD, USA). First, the freeze-dried plasmid was dissolved in sterile water according to the manufacturer’s instructions. Then, 1 µg of plasmid DNA was diluted in 250 µL of cold Opti-MEM and carefully mixed with 4 µL of TurboFectin 8.0 Transfection Reagent (TF81001, ORIGENE, USA). The whole mixture was incubated for 15 min at room temperature, and the cells were transfected at a confluence of 70%. The incubation was carried out for 48 h at 37 °C.

Experiments using an in vitro hypoxia–reoxygenation model were performed in the following way: AC16 cells were exposed to a hypoxic chamber (94.9% N_2_, 5.0% CO_2_, 0.1% O_2_) for 48 h, followed by reoxygenation for one hour under normoxic conditions (5% CO_2_, 95% room air). Control cultures were incubated for the same time period under normoxic conditions. The incubation temperature was 37 °C.

### 2.5. Real-Time RT-PCR

Total RNA was isolated from cardiac tissue, isolated cells, and AC16 cardiomyocytes using peqGold TriFast (peqlab, Biotechnologie GmbH, Burgwedel, Germany) according to the manufacturer’s protocol. cDNA synthesis was performed using SuperScript™ III reverse transcriptase (#18080093, Invitrogen™, Carlsbad, CA, USA) after incubation with 1 U DNase/µg RNA for 15 min at 37 °C. miRNA cDNA synthesis was performed using a microscript microRNA cDNA synthesis kit from Norgen Biotek (ordered via BioCat GmbH, Heidelberg, Germany). Real-time quantitative PCR was performed with the CFX Connect Real-Time PCR Detection System (Bio-Rad Laboratories, Inc., Dreieich, Germany) using iQ™ SYBR^®^ Green Supermix (Bio-Rad Laboratories, Inc., Germany). For information on the primers used, see Appendix A.

### 2.6. SDS-PAGE and Western Blot

Total protein was extracted from cardiac tissue, isolated cells, and AC16 cardiomyocytes using Cell Lysis Buffer (10×) (Cell Signaling, Danvers, MA, USA) according to the manufacturer’s protocol. The homogenate was centrifuged at 14,000× *g* for 10 min at 4 °C, and the supernatant was treated with Laemmli buffer. Protein samples were loaded onto NuPAGE Bis-Tris Precast gels (10%; Life Technology, Darmstadt, Germany) followed by transfer to nitrocellulose membranes. The blots were incubated with an antibody against Roquin-1/2 (MABF288; Life Technology, Darmstadt, Germany) and subsequently with the secondary antibody Goat-Anti-Rat (Jackon ImmunoResearch Europe Ltd., Cambridgeshire, UK).

### 2.7. Statistical Analysis

Data from all groups underwent initial testing for variance (Levene test) and normal distribution (Shapiro–Wilk test). Subsequently, two-sided *T*-tests (2 groups) or a two-sided ANOVA (comparison of multiple groups) was used to compare groups. *p*-Values below 0.05 are indicated as such in the figure legends. Correlation analysis was performed by Pearson correlation using SPSS23. Data are expressed as correlation coefficient β and exact *p*-values.

## 3. Results

### 3.1. Expression of Roquin Isoforms in Rat Heart Tissue

A quantitative comparison of the expression of RC3H1 and RC3H2, the two genes encoding Roquin-1 and Roquin-2, respectively, was performed in isolated adult rat cardiomyocytes and non-myocytes. As indicated in Figure 1A, the primers used in this study to quantify the mRNA expression of Roquin-1 and Roquin-2 have the same primer efficiency. This made it possible to quantitatively compare the expression of both isoforms using PCR thresholds (Figure 1B). Beta-2-Microglobulin (B2M) expression was used as a loading control and showed comparable expression in myocytes and non-myocytes (Figure 1B). Non-myocytes showed a lower threshold and therefore higher expression of RC3H1, whereas in cardiomyocytes RC3H2 showed lower thresholds. The data indicate a cell-specific difference in the expression profile of Roquin isoforms. When the expression in cardiomyocytes is normalized to RC3H2, it becomes clear that the expression of RC3H2 in non-myocytes is marginal (Figure 1C).

### 3.2. Screening for miRNAs That Are Potentially Affected by Roquins

Roquins act as RNA-binding molecules and can directly interact with mature miRNAs and Argonaute 2, which leads to inhibition of miRNA function and blocks Dicer-mediated processing of their precursor forms [17]. To identify potential miRNAs targeted by Roquins, we performed an integrative miRNA-and-mRNA analysis (MMIA). MiRNA expression was initially determined by RNA sequencing and secondarily by RT-PCR in rat ventricular samples taken seven days after the start of reperfusion or from non-ischemic controls [15]. mRNA expression was analyzed by RT-PCR. MiR-34c-3p showed the strongest inverse correlation to RC3H2 and RC3H1 (Figure 2A), but the data could not be confirmed by an alternative approach in which the expression of miR-34c-3p was quantified by RT-PCR instead of RNA sequencing (*RNA-Sequencing: β = −0.616 vs. RT-PCR: β = +0.992*). However, RC3H2 and miR-23b-5p were inversely expressed, regardless of whether the data were generated by RNA sequencing or RT-PCR analysis (Figure 2B,C). Based on this type of analysis, we considered miR-23b-5p as a potential target of Roquin. RC3H2 was down-regulated in all post-ischemic tissues, irrespective of infarct sparing procedures such as pre- and post-conditioning (Figure 2D). In contrast, miR-23b-5p was up-regulated in the same tissues (Figure 2E).

### 3.3. Identification of ZBTB20 as a Potential Target of miR-23b-5p

Once we identified miR-23b-5p as a potential target of Roquin-2, we were interested in identifying potential targets of this miRNA. The data bank TargetScan8.0 suggested ZBTB20 as a potential target of miR-23b-5p. The ZBTB20 gene has three consequential pairing sites for rno-miR-23b-5p and six consequential pairing sites for hsa-miR-23b-5p. If the assumption is correct that miR-23b-5p targets ZBTB20, it must be predicted that this gene is down-regulated in rat heart tissues that showed an increased expression of miR-23b-5p. This assumption was confirmed, as *ZBTB20* is indeed down-regulated in post-ischemic hearts (Figure 3A). As predicted by the relationship between ZBTB20 and cardiac function (Ren et al. 2020; Ren et al. 2024 [18,19]), a strong correlation between *ZBTB20* expression and left ventricular function was found (Figure 3B).

### 3.4. Silencing of Roquin in AC16 Cells

The results on rat heart tissue suggest that ischemic events down-regulate the expression of Roquins, which can then no longer repress the expression of miR-23b-5p. miR-23b-5p subsequently targets the expression of ZBTB20, and down-regulation of this protein translates the signal cascade to a reduced cardiac function. However, these experiments on rat tissues did not validate whether RC3H2 expression correlates with Roquin-2 protein expression. Post-transcriptional modifications may alter such a direct relationship. Moreover, no causal relationship between Roquin expression and miR-23b-5p was shown, only an association of their expression under post-ischemic conditions. To validate the proposed relationship between Roquin, miR-23b-5p, and ZBTB20, expression at the protein level must be confirmed and gene expression must be examined independently of ischemic or hypoxic events. As there is currently no convincing antibody available to detect Roquin in rat tissue, we used AC16 cardiomyocytes, derived from human ventricular tissue, to prove this relationship.

Like rat cardiomyocytes, AC16 cells also constitutively express both Roquin isoforms at the mRNA level (Figure 4A). The application of siRNA against RC3H1 specifically down-regulated the mRNA of Roquin-1, just as the siRNA against RC3H2 exclusively affected the mRNA of Roquin-2 (Figure 4B,C). Moreover, successful down-regulation was also confirmed on the protein levels (Figure 4D–F). Therefore, we used siRNA directed against Roquin isoforms to validate the relationship between Roquin and miR-23b-5p expression.

### 3.5. Relationship Between Roquin-2 Expression and miR-23b-5p in AC16 Cells

Down-regulation of Roquin-2 led to a compensatory up-regulation of miR-23b-5p in normoxic AC16 cells (Figure 5A). Moreover, down-regulation of Roquin-2 also led to a down-regulation of *ZBTB20*, a potential target of miR-23b-5p (Figure 5B). These data suggest that Roquin deficiency and not hypoxia drives the up-regulation of miR-23b-5p and the subsequent down-regulation of *ZBTB20*.

### 3.6. Hypoxia-Dependent Regulation of Roquin in AC16 Cells

In rat tissue, we found a down-regulation of Roquin in post-ischemic tissue, regardless of infarct-sparing procedures. This suggested that ischemia or hypoxia triggers the down-regulation of Roquins and not any events related to reperfusion. This assumption was confirmed in the AC16 cell line, as Roquin-2 was down-regulated at the mRNA and protein level in post-hypoxic cells (Figure 6A–C). As shown above for the silencing of Roquin expression under normoxic conditions, the hypoxia-dependent down-regulation of Roquin was also associated with increased expression of miR-23b-5p and reduced expression of *ZBTB20* (Figure 6D,E).

### 3.7. Effect of Antagomir Treatment Against miR-23b-5p on ZBTB20 Expression

Lastly, we validated the requirement for miR-23b-5p induction as a crucial step for the down-regulation of *ZBTB20*. Again, we down-regulated Roquin expression in AC16 cells using siRNA directed against Roquin-2 and blocked miR-23b-5p activity by antagomir treatment. Antagomir did not affect the down-regulation of Roquin-2 (Figure 7A), but the loss of *ZBTB20* expression was alleviated (Figure 7B).

## 4. Discussion

Our study has revealed a previously undiscovered mechanism by which hypoxia influences ventricular remodeling: hypoxia down-regulates the expression of Roquin in cardiomyocytes, leading to the subsequent up-regulation of miR-23b-5p and ultimately to down-regulation of its target gene *ZBTB20*. This mechanism is important for the subsequent function of the post-ischemic myocardium.

The precise regulation of protein expressions in cells is essential for maintaining cell integrity. Transcription, translation, and protein degradation are primarily involved in maintaining intracellular protein homeostasis. As outlined in the introduction, Roquins belong to an interesting family of proteins that were originally identified as RNA-binding proteins in immune cells and are involved in the regulation of protein expression. They can bind to single-stranded RNA molecules (mRNA and miRNA), thereby reducing transcript concentration and subsequently the translation of RNA into protein [11,17]. They can also affect protein stability via ubiquitination. Roquins can act independently, but in some cases, they can also work together with proteins such as Regnase to regulate target mRNAs. However, studies on other cell types have shown that Regnase can function independently of Roquin [5,20]. Moreover, the expression of Roquin is regulated by MALT1.

Roquins have been characterized as modifiers of immune cell responses, but they are expressed in a broad way in other tissues as well. Our study demonstrates the constitutive expression of Roquin-1 and Roquin-2 in terminally differentiated rat cardiomyocytes and human AC16 cells, a cell line derived from human ventricular cardiomyocytes fused with SV40-transformed fibroblasts. The use of rat terminally differentiated cardiomyocytes and a human cell line has both advantages and disadvantages. The advantages we used in this study are the comparison of cells from two species, the possibility to use antibodies directed against Roquin, and the capability of manipulating cells using siRNA. We also confirmed the expression of Regnase and MALT1 in these cells, but the expression of these proteins was not affected by hypoxia–reoxygenation or ischemia–reperfusion. They were therefore not further investigated in this study.

The role of Roquin in cardiac tissue has not yet been investigated, with the exception of a contribution to T-cell-dependent dysregulation associated with cardiac inflammation [5]. In contrast to this study, our data clearly show that Roquin is expressed in non-immune heart cells, such as cardiomyocytes. Furthermore, we observed a down-regulation of Roquin in post-ischemic rat hearts and post-hypoxic human cardiomyocytes. Interestingly, miR-223 was previously identified as an miRNA that triggers the degradation of Roquin and can also be used as a biomarker after myocardial infarction [21]. Thus, hypoxia may induce the expression of miR-223, which then triggers the hypoxia-dependent degradation of Roquin, but this requires future studies to identify the molecular mechanism more precisely. Our data show a comparative down-regulation of Roquin mRNA and protein expression, suggesting that regulation occurs on the transcriptional level.

We investigated via RNA sequencing the expression of multiple miRNAs in the rat heart under non-ischemic conditions and seven days after ischemia–reperfusion with and without infarct-sparing procedures such as pre- and post-conditioning [15]. We then correlated the expression of these miRNAs with that of Roquin mRNA to identify potential targets of Roquin. Overall, we have indeed identified several miRNAs that show a strong inverse correlation with Roquin expression, suggesting that these miRNAs are also potential targets of Roquin. It is important to validate the expression of these miRNAs with two independent methods—RNA sequencing and RT-PCR. After validation of miRNA expression with these two independent methods, we identified miR-23b-5p as a potential target of Roquins in cardiac cells. These correlations were subsequently confirmed in AC16 cells with Roquin silencing under normoxic conditions. Our data show that it is indeed Roquin deficiency, and not hypoxia itself, that triggers the inverse regulation of miR-23b-5p.

Next, we used the TargetScan8.0 data bank to identify potential targets of miR-23b-5p. Based on conserved sequences in both rat and human gene sequences, *ZBTB20* was identified as a potential down-stream target of miR-23b-5p. A deficiency of ZBTB20 is associated with cardiac dysfunction and a lack of cardiac reserve and exercise capacity in mice [18,19]. Mechanistically, ZBTB20 is required for phospholamban expression, and its deficiency leads to SERCA2a overactivation and increased Ca^2+^ loading of the sarcoplasmic reticulum (SR). Here, we show that hypoxia-dependent down-regulation of Roquin increased the expression of miR-23b-5p. This is subsequently linked to a reduction in *ZBTB20* expression, and this is associated with reduced left ventricular function. Furthermore, siRNA-induced down-regulation of Roquin and subsequent induction of miR-23b-5p did not lead to a reduction in *ZBTB20* expression when antagomir against miR-23b-5p was administered simultaneously.

As in all studies, our study has some limitations that cannot properly be addressed in this manuscript but require future attention. The lack of a proper antibody to detect Roquin-2 in rat tissue means that we could confirm the co-regulation of roquin mRNA and protein only in AC16 cells. However, we have no reason to assume that this will be different in rats. The different expression profile between Roquin-1 and Roquin-2 in rat tissue and AC16 cells requires attention, i.e., the role of Roquin-1 remains unclear, as we could not find any major difference between either isoform in this study. Furthermore, we believe that no differences occur between the Roquin expression in males and females, but this was not really checked in this study. From the data so far, there is no evidence to assume such a difference. Finally, the in vitro part has the limitation that the cells were not exposed to physiological pO_2_ levels, as they were cultured in a standard incubator using room air and 5% pCO_2_. Thus, the challenge to hypoxia and reoxygenation does not exactly mimic the in vivo situation of ischemia–reperfusion. Nevertheless, this limitation is overbalanced by the finding of a similar type of regulation between in vivo and in vitro conditions and the ability to use siRNA and antagomirs to manipulate Roquin expression.

In summary, this is the first study to demonstrate the constitutive expression of Roquin isoforms in cardiomyocytes, their down-regulation in post-hypoxic tissues, an effect on miR-23b-5p and thus on *ZBTB20*, and finally an association with reduced cardiac function (Figure 8). Overall, the mechanisms identified may contribute to our understanding of why cardioprotective measures such as pre-conditioning and post-conditioning alone cannot completely prevent post-ischemic cardiac dysfunction, as they do not address hypoxia-induced alterations. Furthermore, our study suggests that the development of new techniques aimed to increase the expression of Roquins could lead to improved recovery after myocardial infarction.

## Figures and Tables

**Figure 1 cells-14-01748-f001:**
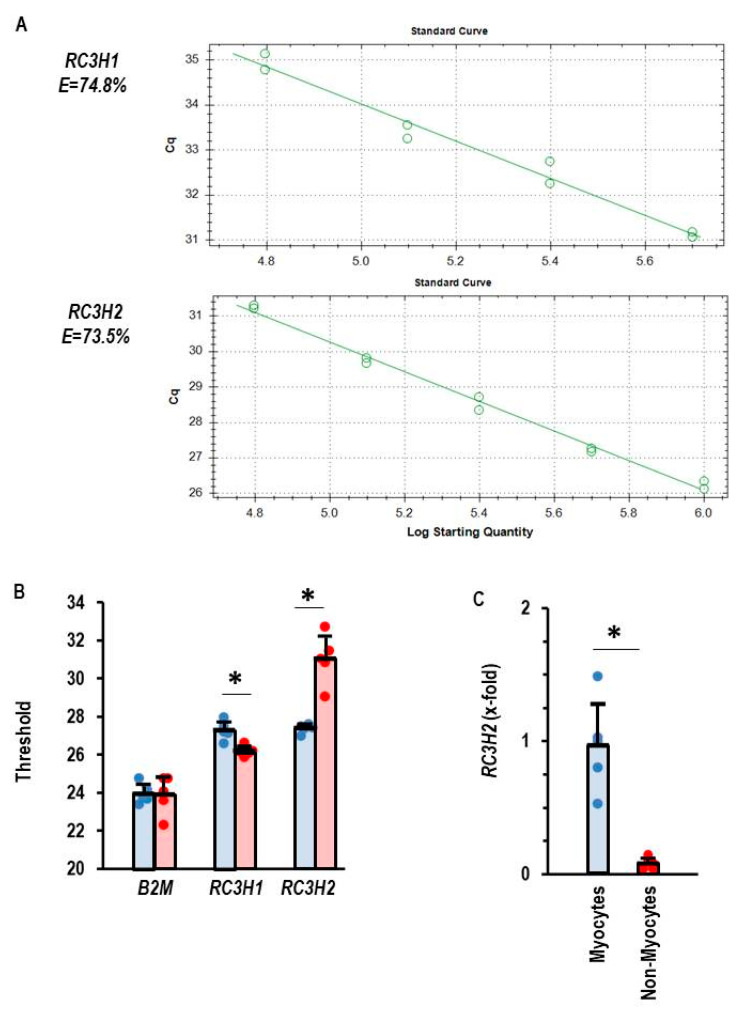
Expression of Roquin mRNA in rat heart tissue. (**A**) Primer efficiency for *RC3H1* and *RC3H2*. (**B**) Thresholds for PCR analysis for the house-keeping gene beta-2-microglubulin (*B2M*), *RC3H1*, and *RC3H2*. Myocytes in blue and non-myocytes in red. (**C**) Normalized expression of *RC3H2* in myocytes and non-myocytes with mean expression of *RC3H2* in cardiomyocytes set as 1. Data are given as means + S.D. (in black) and individual data points. *: *p* < 0.05 between myocytes and non-myocytes. *T*-test with Levene test for variance testing and Shapiro–Wilk test for normal distribution.

**Figure 2 cells-14-01748-f002:**
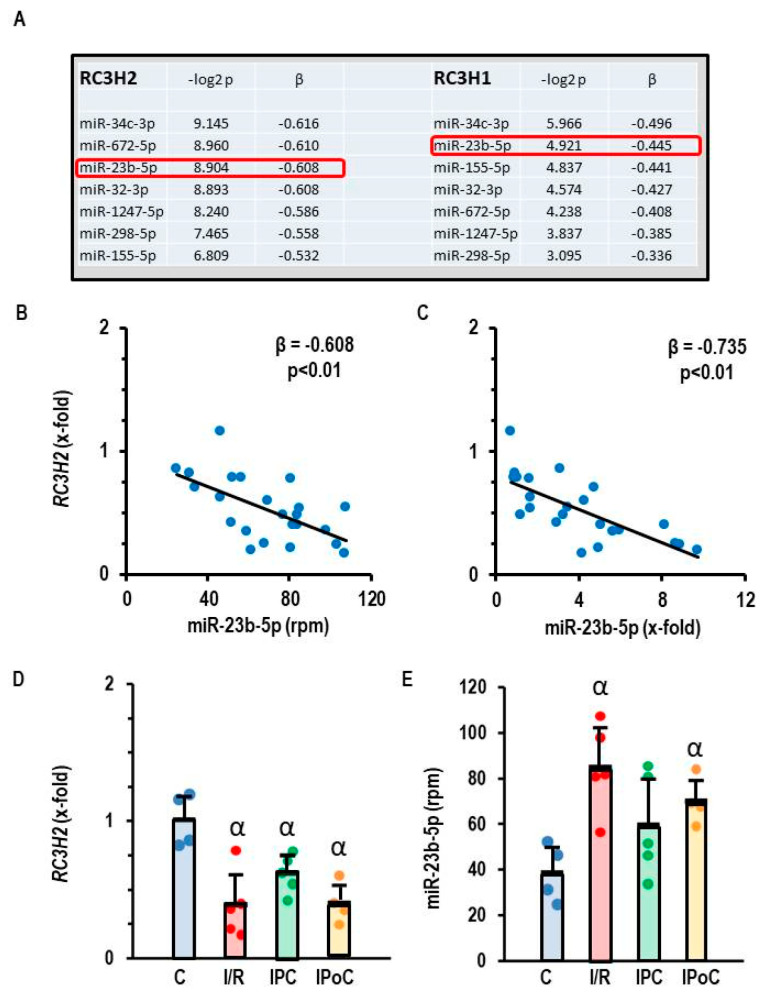
Correlation between *RC3H2* and *RC3H1* expression and the expression of miR-23b-5p in rat heart tissue. (**A**) Pearson correlation with *p*-values and β coefficient. (**B**,**C**) Correlation between the expression of *RC3H2* and miR-23b-5p quantified by RNA sequencing, expressed as rpm (**B**) or by RT-PCR, expressed as fold change of non-ischemic controls. (**D**) Expression of *RC3H2* in the left ventricle of samples from control rats (C), 45 min ischemia and 7-day reperfusion (I/R), 45 min ischemia with pre-conditioning and 7-day reperfusion (IPC), and 45 min ischemia with subsequent pre-conditioning and 7-day reperfusion (IPoC). (**E**) Expression of miR-23b-5p (expressed as reads per million, rpm) in the left ventricle of samples from control rats (C), 45 min ischemia and 7-day reperfusion (I/R), 45 min ischemia with pre-conditioning and 7-day reperfusion (IPC), and 45 min ischemia with subsequent pre-conditioning and 7-day reperfusion (IPoC). Data are given as means + S.D. and individual data points. α: *p* < 0.05 different from control rats (C). Two-sided ANOVA with Student–Newman–Keuls post hoc analysis. Exact *p*-values are given for correlation analysis.

**Figure 3 cells-14-01748-f003:**
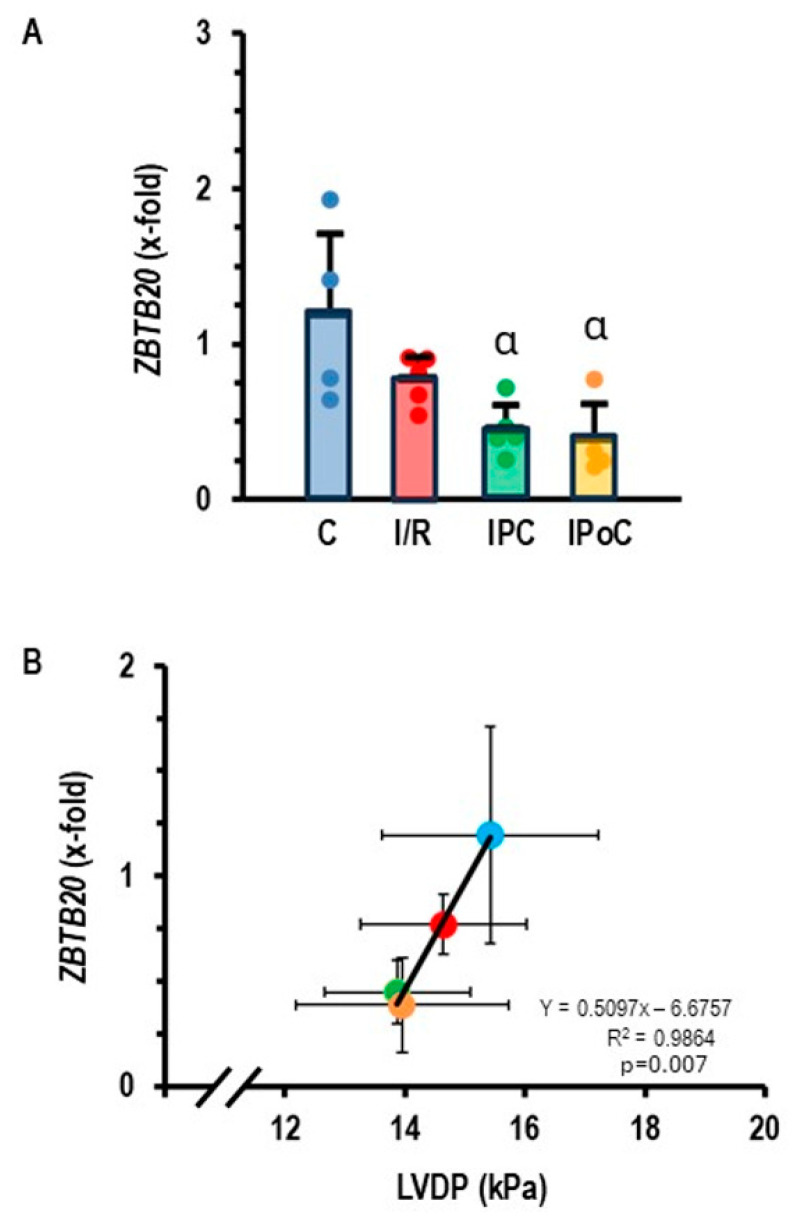
(**A**) Expression of *ZBTB20* in the left ventricle of samples from control rats (C), 45 min ischemia and 7-day reperfusion (I/R), 45 min ischemia with pre-conditioning and 7-day reperfusion (IPC), and 45 min ischemia with subsequent pre-conditioning and 7-day reperfusion (IPoC). Data are normalized to the expression of non-ischemic controls. Data are given as means + S.D. (in black) and data points. α: *p* < 0.05 versus control. Two-sided ANOVA with Student–Newman–Keuls post hoc analysis. (**B**) Correlation between *ZBTB20* expression and left ventricular developed pressure (LVDP). Exact *p*-value is given for the Pearson correlation analysis.

**Figure 4 cells-14-01748-f004:**
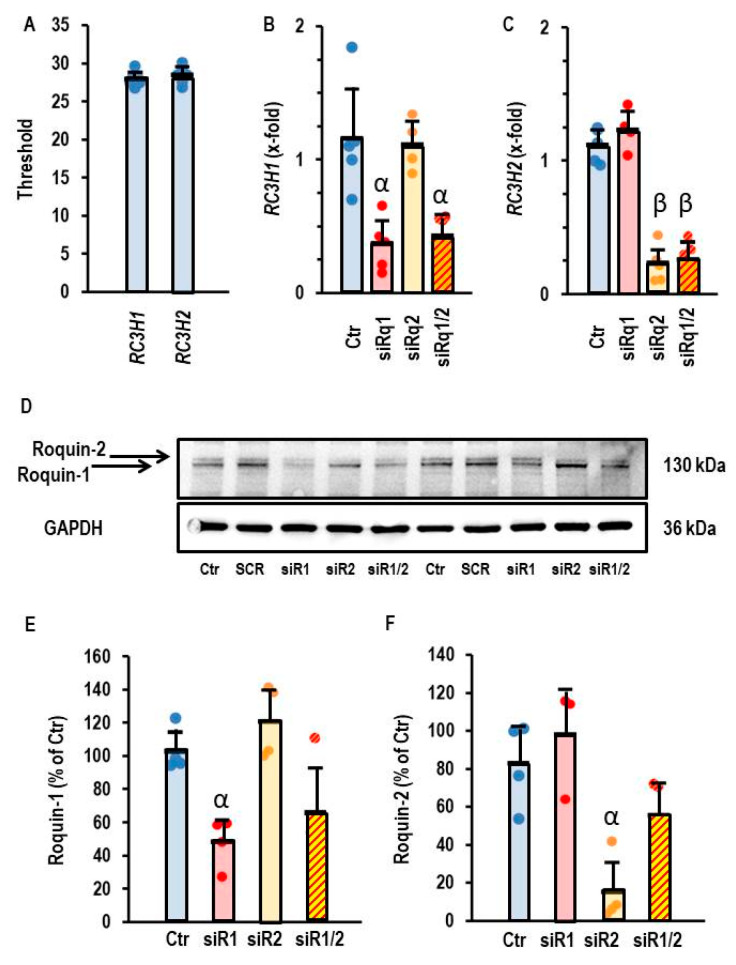
Effect of siRNA directed against RC3H1 and RC3H2 in AC16 cells. (**A**) Thresholds of PCR analysis for both genes. (**B**) Expression of RC3H1 in AC16 cells after incubation with siRNA directed against both Roquin isoforms. (**C**) Expression of RC3H2 in AC16 cells after incubation with siRNA directed both against Roquin isoforms. Data are given as means + S.D. (in black) and individual data points. β: *p* < 0.05 between siR2 vs. Ctr. and siR1. α: *p* < 0.05 for siR1 vs. Ctr and siR2. Two-way ANOVA with Student–Newman–Keuls post hoc analysis. (**D**) Western blot showing the expression of both Roquin isoforms in cells that were or were not (Ctr) transfected with siRNA against Roquin. (**E**,**F**) Quantification of the Western blot shown in (**D**) with α, *p* < 0.05 vs. Ctr.

**Figure 5 cells-14-01748-f005:**
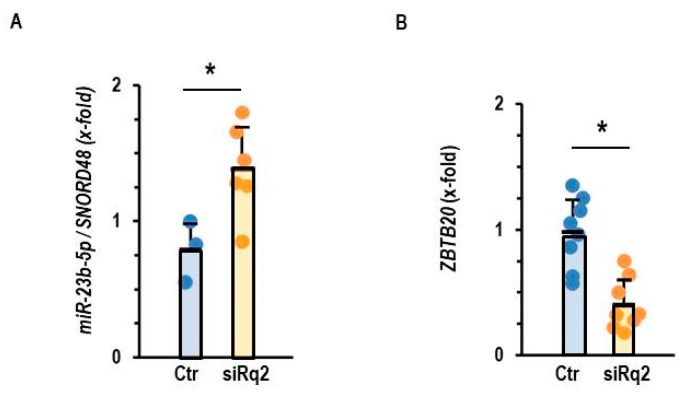
Effect of Roquin silencing on the expression of miR-23b-5p and *ZBTB20*. (**A**) Expression of miR-23b-5p in AC16 cells after incubation with siRNA directed against Roquin 2. (**B**) Expression of *ZBTB20* in AC16 cells after incubation with siRNA directed against Roquin-2. Data are given as means + S.D. (in black) and individual data points. *: *p* < 0.05 vs. Ctr. with unpaired *T*-tests.

**Figure 6 cells-14-01748-f006:**
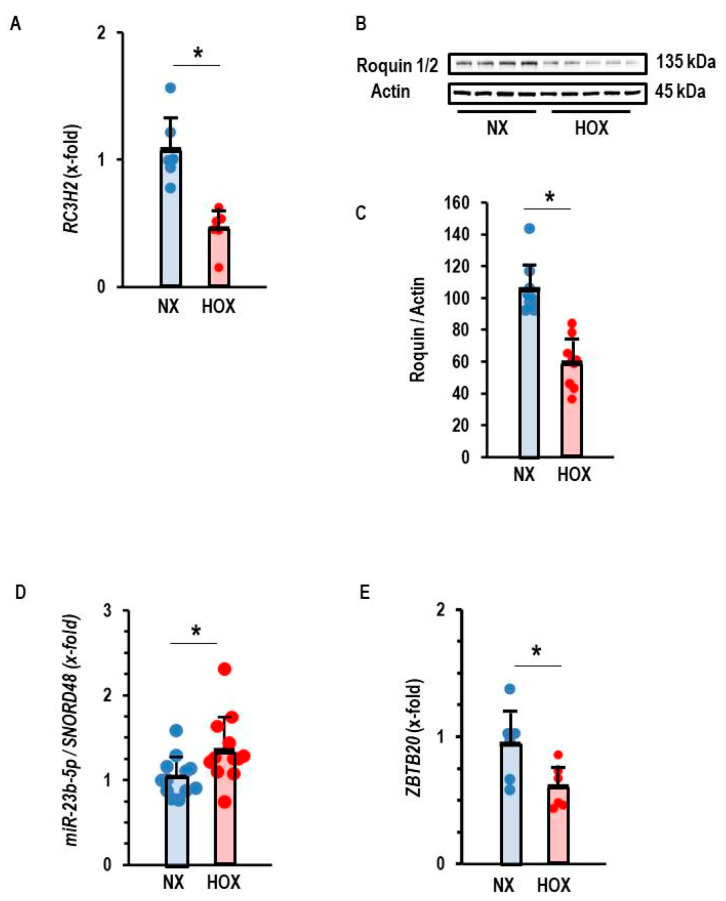
Effect of hypoxia (Hox) on the expression of Roquin. (**A**) Expression of RC3H2. Data are normalized to normoxic controls (Nx). (**B**) Western blot indicating the effect of hypoxia (Hox) on the expression of Roquin and quantification of the Western blot in (**C**) with *p* < 0.05 (unpaired *T*-test with Levene test for variance testing and Shapiro–Wilk test for normal distribution). (**D**) Expression of miR-23b-5p normalized to *SNORD48* as loading control. (**E**) Expression of *ZBTB20*. Data are given as means + S.D. (in black) and individual data points. *: *p* < 0.05 vs. Ctr. unpaired *T*-test with Levene test for variance testing and Shapiro–Wilk test for normal distribution.

**Figure 7 cells-14-01748-f007:**
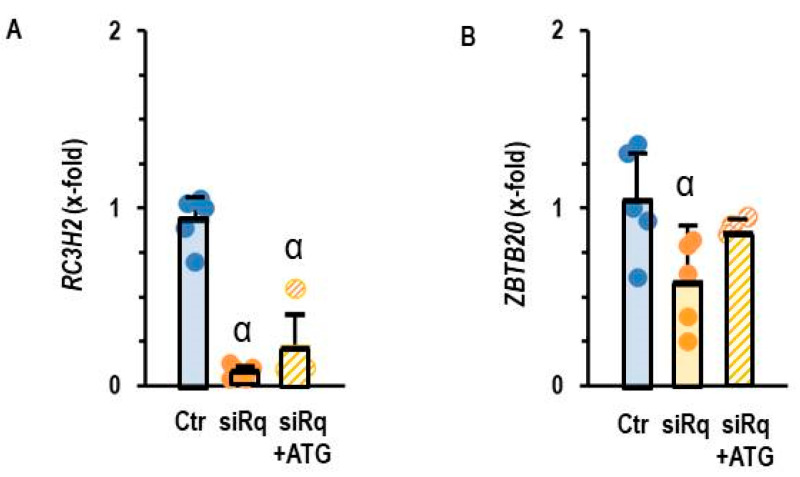
Expression of (**A**) *RC3H2* and (**B**) *ZBTB20* in AC16 cells with silencing of Roquin and antagomir treatment against miR23b-5p (ATG). Data are given as means + S.D. (in black) and individual data points. α: *p* < 0.05 versus control. Two-sided ANOVA with Student–Newman–Keuls post hoc analysis.

**Figure 8 cells-14-01748-f008:**
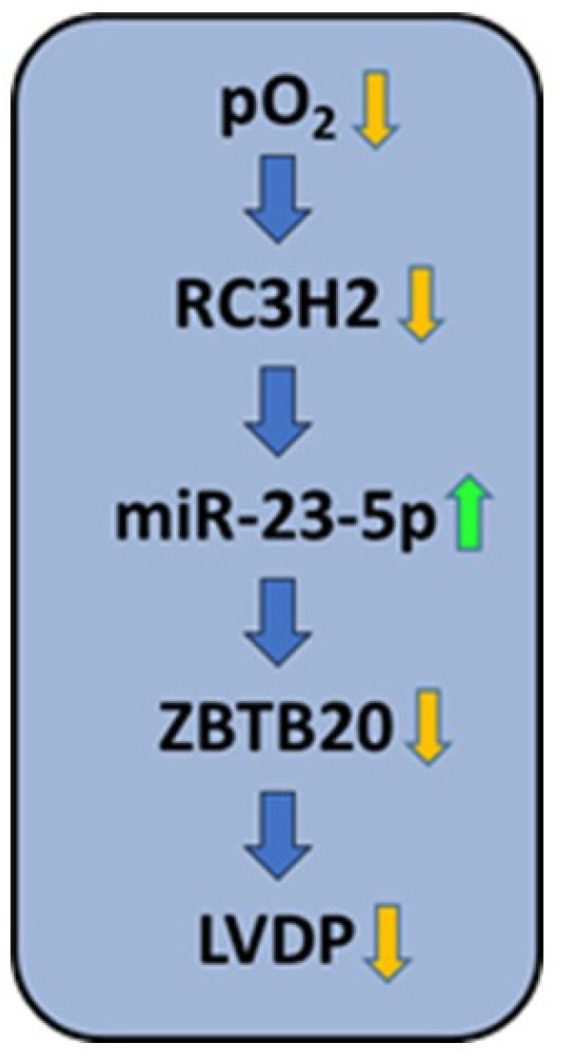
Summary of the findings: Hypoxia down-regulates the expression of *RC3H2* (Roquin-2), as demonstrated by exposing AC16 cells to hypoxia and rat hearts to ischemia (Figure 2D and Figure 6A–C). Down-regulation of Roquin-2, as shown at the mRNA and protein level, increases the stability of miR-23b-5p (Figure 2A–C, Figure 5A, Figure 6D and Figure 7A). Increased expression of miR-23b-5p subsequently led to reduced expression of ZBTB20 (Figure 5B, Figure 6E and Figure 7B), and this is associated with decreased function (left ventricular developed pressure, LVDP; Figure 3B).

## Data Availability

Original data can be requested from the corresponding author.

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
