# Peer review of "Roquin Modulates Cardiac Post-Infarct Remodeling via microRNA Stability Control"

_cells, 2025, doi:10.3390/cells14221748_

Round 1

Reviewer 1 Report

Comments and Suggestions for Authors

This manuscript addresses a potentially very interesting and novel topic — the role of Roquins in the context of myocardial ischemia/reperfusion (I/R). To the best of my knowledge, no peer-reviewed studies have yet explored this aspect, and therefore the Authors’ findings could represent an important contribution to the field. The evidence demonstrating the presence of Roquins in post-I/R myocardial tissue appears robust and convincing. However, the section dealing with the regulation of Roquins under hypoxic conditions in cultured cells is comparatively weaker and requires significant clarification.

Regarding the cardiac tissue analyses, there is an issue that needs to be addressed. It seems that the Authors used tissue samples from previous studies, which is perfectly acceptable for exploratory research. However, in line 82, they refer to female Wistar-Hannover rats, whereas in line 89, they mention male Wistar rats. The potential mismatch in sex and strain raises concerns about sample heterogeneity and comparability. The Authors should clarify whether these discrepancies are typographical errors or reflect actual differences in the experimental material.

Concerning the in vitro experiments, the Authors should clearly specify: (A) the oxygen concentration (%O₂) used to define hypoxia, (B) the duration of exposure, and (C) whether reoxygenation was applied prior to cell harvesting. If reoxygenation was performed, the experimental setting models I/R; otherwise, it represents ischemia only. It should also be recalled that 21% O₂ does not correspond to physiological normoxia for myocardial cells, but rather to hyperoxia. This distinction is not merely semantic — cellular responses to hypoxia and hyperoxia are often non-linear and may exhibit a U-shaped relationship. Without such clarifications, the conclusions regarding the role of hypoxia in altering Roquins and ZBTB20 remain unsubstantiated.

If confirmed and more clearly contextualized, the findings presented in this study could indeed become of considerable interest. The conceptual summary provided in Figure 8, however, is overly simplistic; a more detailed and integrated schematic would help the reader better understand the proposed mechanisms.

Minor Comments

Line 169: Please define AGO2 upon first mention and explain it.

Figure 3B: The regression line would be more convincing if accompanied by 95% confidence limits. Additionally, consider using different colors or symbols to match individual points with those shown in Figure 3A.

Reviewer 2 Report

Comments and Suggestions for Authors

The study 1) identifies a previously undiscovered mechanism where hypoxia down-regulates Roquin (both Roquin-1 and Roquin-2) in cardiomyocytes, leading to increased stability of miR-23b-5p and subsequent down-regulation of ZBTB20, impacting post-infarct remodeling; 2) utilizes an integrative miRNA-mRNA analysis (MMIA) combined with experimental validation (e.g., siRNA, antagomir) in both rat heart tissue and human AC16 cardiomyocytes, strengthening the findings; 3) demonstrates a correlation between ZBTB20 expression and left ventricular function, suggesting potential therapeutic targets for improving recovery after myocardial infarction; 4) includes detailed methods (e.g., real-time RT-PCR, Western blot, statistical analysis) and ethical statements, enhancing reproducibility. By reading throughout, the areas needed to be Improvement:

  1. Clarity and Consistency: Some sections (e.g., Abstract, Introduction) contain minor typographical errors (e.g., "identifing" instead of "identifying," "si-lencing" instead of "silencing"). Proofreading is recommended.
  2. Data Validation: The initial inverse correlation of miR-34c-3p with Roquin isoforms was not confirmed by RT-PCR, suggesting the need for further validation of other potential miRNA targets identified in the MMIA.
  3. Mechanistic Depth: The role of miR-223 in Roquin degradation under hypoxia is mentioned but not explored, which could be a limitation. Future studies to confirm this pathway are suggested.
  4. Figure Legends: Some figure legends (e.g., Figure 2) could be more detailed to clarify statistical tests and abbreviations (e.g., rpm) for non-specialist readers.
  5. Discussion Expansion: The discussion could better address limitations, such as the lack of direct evidence for Roquin protein levels in rat tissue due to antibody issues, and how this might affect conclusions.
Comments on the Quality of English Language

Clarity and Consistency: Some sections (e.g., Abstract, Introduction) contain minor typographical errors (e.g., "identifing" instead of "identifying," "si-lencing" instead of "silencing"). Proofreading is recommended.

Round 2

Reviewer 1 Report

Comments and Suggestions for Authors

I sincerely thank the Authors for answering more than satisfactorily my concerns.

As for point 1 (cardiac tissue analyses), now I agree with them, but I suggest adding a sentence or two addressing this issue as a limitation of the study in the text. Indeed, I noticed that a new paragraph dealing with some limitations has been already added.

As for the cell cultivation issue, I also encourage adding a few sentences addressing the possible non-linearity of the cell responses to hypoxia and hyperoxia.

Finally, I really like the new figure, but the uploaded version still reports the old one.
